# In Silico Analysis of Peptide-Based Derivatives Containing Bifunctional Warheads Engaging Prime and Non-Prime Subsites to Covalent Binding SARS-CoV-2 Main Protease (M$^{pro}$)

Simone Brogi [1,*] , Sara Rossi [2] , Roberta Ibba [2] , Stefania Butini [2] , Vincenzo Calderone [1] , Giuseppe Campiani [2] and Sandra Gemma [2,*]

1  Department of Pharmacy, University of Pisa, Via Bonanno 6, 56126 Pisa, Italy; vincenzo.calderone@unipi.it
2  Department of Biotechnology, Chemistry and Pharmacy, DoE Department of Excellence 2018–2022, University of Siena, Via Aldo Moro 2, 53100 Siena, Italy; rossi115@student.unisi.it (S.R.); roberta.ibba@unisi.it (R.I.); butini3@unisi.it (S.B.); campiani@unisi.it (G.C.)
*  Correspondence: simone.brogi@unipi.it (S.B.); gemma@unisi.it (S.G.)

**Abstract:** Despite the progress of therapeutic approaches for treating COVID-19 infection, the interest in developing effective antiviral agents is still high, due to the possibility of the insurgence of viable SARS-CoV-2-resistant strains. Accordingly, in this article, we describe a computational protocol for identifying possible SARS-CoV-2 M$^{pro}$ covalent inhibitors. Combining several in silico techniques, we evaluated the potential of the peptide-based scaffold with different warheads as a significant alternative to nitriles and aldehyde electrophilic groups. We rationally designed four potential inhibitors containing difluorstatone and a Michael acceptor as warheads. In silico analysis, based on molecular docking, covalent docking, molecular dynamics simulation, and FEP, indicated that the conceived compounds could act as covalent inhibitors of M$^{pro}$ and that the investigated warheads can be used for designing covalent inhibitors against serine or cysteine proteases such as SARS-CoV-2 M$^{pro}$. Our work enriches the knowledge on SARS-CoV-2 M$^{pro}$, providing a novel potential strategy for its inhibition, paving the way for the development of effective antivirals.

**Keywords:** SARS-CoV-2; main protease (M$^{pro}$); computer-aided drug design; molecular docking; molecular dynamics

## 1. Introduction

Severe acute respiratory syndrome coronavirus-2, widely known as SARS-CoV-2, is the etiological agent of COVID-19 that caused several epidemic outbreaks since its first appearance in 2019 in the city of Wuhan, China [1]. Since then, rapid vaccination campaigns have been implemented at the global level to protect the population from the most severe symptoms. Moreover, very recently novel antivirals such as molnupiravir and paxlovid (PF-07321332 + ritonavir) have been added to the COVID-19 therapeutic armamentarium to treat the infection in patients with high risk of severe symptoms [2–5]. However, the research of effective antivirals remains a priority, both for the current and future pandemics. SARS-CoV-2 belongs to the *Coronaviridae* subfamily which is composed of alpha-, beta-, gamma-, and delta-CoVs [6]. The SARS-CoV-2 genome comprises approximately 30,000 nucleotides that feature genes for the production of nonstructural proteins (enzymes required for viral transcription and replication) and structural proteins. The life cycle of the virus begins when the spike glycoprotein (S) binds the host receptor, which, in the case of SARS-CoV-2, is the ACE2 enzyme. This interaction determines the fusion of the cell membrane with the viral one and allows the entry of the virus inside the cell. Once inside the host cell, the virus disassembles to release the nucleocapsid and viral RNA into the cytoplasm. Afterward, translation of the ORF1a/b takes place to form the large replicase

polyprotein 1a (pp1a) and pp2ab, and the replication of genomic RNA occurs. The pp1ab polyprotein is processed by two viral proteases, 3-chymotrypsin-like protease (3CL^pro or M^pro) and papain-like protease enzyme (PL^pro), to release nonstructural proteins such as RNA-dependent RNA polymerase and helicase, which are involved in viral transcription and replication and the structural proteins that will form the new viral particles. The virion is assembled in the endoplasmic reticulum and Golgi and is finally released into the extracellular compartment by exocytosis.

The M^pro enzyme as a target for developing new antiviral drugs: The M^pro enzyme is one of the best characterized and validated as a drug target among those known for coronaviruses [7]. Together with PL^pro, it is essential for the maturation of the polyprotein, which is translated from viral RNA and cleaved by the proteases. The M^pro enzyme operates no less than 11 hydrolytic breaks on the polyprotein 1ab in correspondence with specific recognition sequences. Most of the cleavage sites hold Leu-Gln ↓ (Ser, Ala, Gly) as the recognition sequence. Hence, M^pro inhibition blocks viral replication [8].

Among the three antivirals currently approved for the treatment of SARS-CoV-2 infection, PF-07321332 (**1**, Figure 1) is an M^pro inhibitor [9,10]. Its structure is characterized by a peptidomimetic scaffold bearing a nitrile moiety as an electrophilic warhead.

**Figure 1.** Chemical structures of reported SARS-CoV-2 M^pro inhibitors (**1–3**) showing different electrophilic warheads.

In general, the design of serine and cysteine protease inhibitors involves the insertion of electrophilic groups (warheads) that are reversibly or irreversibly attacked by nucleophilic serine or cysteine catalytic residues to form covalent adducts. The selectivity of the inhibitors is guaranteed by warhead flanking moieties able to specifically interact with the subsites of the enzyme mimicking the endogenous peptidic substrates. For these reasons, serine/cysteine protease inhibitors are usually characterized by a peptidic or peptidomimetic structure, albeit nonpeptidic M^pro inhibitors have also been reported [11]. Compounds **2** and **3** in Figure 1 are examples of different mono- or bifunctional warheads [8,12].

The electrophilic warhead plays a critical role in the development of M^pro inhibitors since it has to be characterized by sufficient reactivity to react with active-site residues, but stable enough not to engage in unwanted and aspecific reactions with other nucleophiles;

it should be readily accommodated inside the active site and be able to appropriately orientate the flanking moieties toward the enzyme subsites. In order to better understand the potential binding mode of electrophilic warheads and the role of the affinity of flanking substituents, it is important to implement appropriate computational protocols able to fully elucidate the parameters involved in covalent and noncovalent interactions. Here, we report an in silico protocol aimed at investigating the potential binding mode and reactivity of two different bifunctional warheads (compounds **4**–**7**, Figure 2). We chose to investigate in silico bifunctional electrophilic moieties that can be functionalized at both sides in order to engage with residues at both the prime and nonprime subsites of the enzyme or to be exploited to modulate drug-like properties. In particular, the difluorostatone-based warhead has been demonstrated by us and others to be able to engage in reversible covalent interactions with different serine proteases [13–15].

**Figure 2.** Chemical structures of potential inhibitors of SARS-CoV-2 M$^{pro}$ (**4**–**7**) reported in this study.

On the other hand, Michael-based acceptor electrophilic moieties have been previously reported as alternatives to nitriles and aldehyde warheads. Starting from inhibitor **3**, the structural models **4**–**6** used for our computational investigation were designed by keeping constant residues P1–P3 and replacing the Michael acceptor group with a difluorostatone moiety. In particular, we wanted to verify if, in our computational protocol, compound **6** could be potentially able to form H-bond interactions inside the S2 subsite, similarly to what is described for reference inhibitor **2**. Here, we report a preliminary in silico investigation aimed at assessing the potential binding mode of the difluorostatone/aza-Michael moieties.

Thus, we conducted an extensive computational investigation based on molecular docking, molecular dynamics, and covalent docking approaches for determining the potential of the conceived compounds in inhibiting SARS-CoV-2 M$^{pro}$.

## 2. Materials and Methods

### 2.1. Computational Details

#### 2.1.1. Protein and Ligand Preparation

Peptide-based derivatives were built in Maestro Molecular Modeling Suite (Maestro release 2020-3, Schrödinger, LLC, New York, NY, USA, 2020) using the available drawing tools as described [13,16]. Energy minimization was performed using the MacroModel application with the OPLS-2005 force field [17]. The resulting compounds were treated by LigPrep software (LigPrep release 2020, Schrödinger, LLC, New York, NY, USA, 2020) to provide the most probable ionization state at physiological pH (7.4 ± 0.2). To simulate solvent effects, the GBSA model was used with "no cutoff" for nonbonded interactions. The

PRCG method (5000 maximum iterations and threshold for gradient convergence = 0.001) was used to minimize the potential energy. The experimental structure of the SARS-CoV-2 $M^{pro}$ enzyme was downloaded from the Protein Data Bank (PDB ID: 6Y2G [12]; crystal structure of $M^{pro}$ in complex with α-ketoamide-based covalent inhibitor) and imported into Maestro Suite 2020. The first step was to break the covalent bond between C145 and the α-ketoamide derivative to restore the native arrangement of the enzyme. Next, to refine the structure, we applied the Protein Preparation Wizard protocol available in Maestro for performing various computational steps to (1) add hydrogens; (2) optimize the orientation of hydroxyl groups of residues, Asn and Gln, and the protonation state of His; and (3) perform a constrained minimization refinement using the *impref* utility. At first, the protein was preprocessed by adding all hydrogen atoms to the structure, assigning bond orders, creating disulfide bonds, and filling missing sidechains and loops. To optimize the hydrogen bond network, His tautomers and ionization states were predicted, 180° rotations of the terminal angle of Asn, Gln, and His residues were assigned, and hydrogen atoms of the hydroxyl and thiol groups of residues were sampled. Finally, a restrained minimization was performed using the Impact Refinement (*impref*) module, employing the OPLS3 force field to optimize the geometry and minimize the energy of the protein. The minimization was terminated when the energy converged or the root-mean-square deviation (RMSD) reached a maximum cutoff of 0.30 Å.

### 2.1.2. Molecular Docking

Glide software (Glide release 2020, Schrödinger, LLC, New York, NY, USA, 2020) employing the SP scoring function was used to perform all docking studies conducted in this work [18]. The energy grid for docking was prepared using the default value of the protein atom-scaling factor (1.0 Å), with a cubic box centered on the crystallized ligand. The docked poses considered for the post-docking minimization step were 1000.

To improve the quality of the investigation, we also evaluated the ligand-binding energies from the complexes derived by the docking calculation. For this purpose, the Prime/MM-GBSA method, available in Prime software (Prime release 2020, Schrödinger, LLC, New York, NY, USA, 2020), was used. This technique computes the variation between the free and the complex states of both the ligand and enzyme after energy minimization [19,20].

### 2.1.3. Molecular Dynamics

Desmond 5.6 academic version, provided by D. E. Shaw Research ("DESRES"), was utilized to perform MD simulation experiments via the Maestro graphical interface (Desmond Molecular Dynamics System, version 5.6, D. E. Shaw Research, New York, NY, USA, 2018. Maestro-Desmond Interoperability Tools, Schrödinger, New York, NY, USA, 2018). MD was performed using the Compute Unified Device Architecture (CUDA) API [21] on two NVIDIA GPUs. The Desmond system builder available via Maestro was employed for solvating the complexes derived from the docking studies into an orthorhombic box filled with water, simulated by the TIP3P model [22,23]. The OPLS force field [17] was used for MD calculations as reported [23–25]. To simulate the physiological concentration of monovalent ions, we added $Na^+$ and $Cl^-$ ions to obtain a final salt concentration of 0.15 M. Constant temperature (300 K) and pressure (1.01325 bar) were employed with the NPT (constant number of particles, pressure, and temperature) as the ensemble class. The RESPA integrator [26] was used to integrate the equations of motion, with an inner time step of 2.0 fs for bonded and nonbonded interactions within the short-range cutoff. Nose–Hoover thermostats [27] were used to keep the constant simulation temperature, and the Martyna–Tobias–Klein method [28] was applied to control the pressure. Long-range electrostatic interactions were calculated by the particle-mesh Ewald method (PME) [29]. The cutoff for van der Waals and short-range electrostatic interactions was set at 9.0 Å. The equilibration of the system was performed using the default protocol provided in Desmond, which consists of a series of restrained minimization and MD simulations applied to slowly relax the

system. Consequently, one individual trajectory for each complex of 100 ns was calculated. The trajectory files were analyzed by MD analysis tools available in Maestro. The same applications were used for generating all plots regarding MD simulations presented in this article. Therefore, the RMSD was calculated using the following equation:

$$RMSD_x = \sqrt{\frac{1}{N} \sum_{i=1}^{N} (r_i'(t_x) - r_i\ (t_{ref}))^2}$$

where $RMSD_x$ refers to the calculation for a frame $x$; $N$ is the number of atoms in the atom selection; $t_{ref}$ is the reference time (typically the first frame is used as the reference, and it is regarded as time $t = 0$); and $r'$ is the position of the selected atoms in frame x, after superimposing on the reference frame, where frame x is recorded at time $t_x$. The procedure is repeated for every frame in the simulation trajectory. Regarding the root-mean-square fluctuation (*RMSF*), the following equation was used for the calculation:

$$RMSF_i = \sqrt{\frac{1}{T} \sum_{t=1}^{T} <(r_i'(t) - r_i\ (t_{ref}))^2>}$$

where $RMSF_i$ refers to a generic residue $i$, $T$ is the trajectory time over which the *RMSF* is calculated, $t_{ref}$ is the reference time, $r_i$ is the position of residue I, $r'$ is the position of atoms in residue $i$ after superposition on the reference, and the angle brackets indicate that the average of the square distance is taken over the selection of atoms in the residue.

Free-energy perturbation (FEP) was performed using the FEP module available in the Desmond package using the complexes obtained by docking calculations, employing the default setting of the FEP protocol. The simulation was split into 12 $\lambda$-windows, with replica exchange attempted every 1.2 ps.

### 2.1.4. Covalent Docking

Covalent docking studies were executed in Maestro Suite 2020 applying the Covalent Docking protocol (CovDock) [30] as previously reported by us [14,31]. The algorithm utilizes both the Glide docking algorithm and Prime structure refinement. The CovDock application considers custom reactions enclosed in a list of possible covalent reactions (implemented in the software) using the SMARTS pattern. In this way, it is possible to automatically recognize the reactive residue and the portion of the ligand that are involved in the reaction. If the desired reaction is not present in that list, it is possible to write the reaction that involves the correct atoms. In this study, since the desired reaction considering the difluorostatone derivatives was not present in the list of reactions provided by CovDock, the reaction of the SMARTS pattern was customized [CC(C)=O] to obtain a reliable reaction for the compounds. Instead, the reaction involving a Michael acceptor is present in the reaction list. To start the calculation, the reactive residue of the receptor was selected (C145) and matched to the one defined in the custom chemistry file to specify the reaction type. The grid center was positioned at the centroid of the selected docked ligand, and the size of the grid box was automatically determined. No constraints were used, and the pose prediction option was selected for obtaining more accurate output results. Following the docking procedure, the obtained poses were filtered using default parameters, and the scoring option MM/GBSA was selected.

### 2.1.5. Physicochemical Properties Evaluation

QikProp (QikProp release 2020, Schrödinger, LLC, New York, NY, USA, 2020) was used for assessing logP and logS, while the possible pan-assay interference compounds (PAINS) issue was evaluated employing the online server FAFDrugs4 (https://fafdrugs4 .rpbs.univ-paris-diderot.fr/ accessed on 25 February, 2022).

## 3. Results and Discussion

### 3.1. Molecular Docking Studies

In order to assess the tendency of our conceived compounds, reported in Figure 2, to bind the SARS-CoV-2 M^pro enzyme, we conducted a series of in silico experiments mainly based on molecular docking and molecular dynamics (Table 1). The protein (PDB ID: 6Y2G) and the ligands prepared as reported in the Materials and Methods section were docked into the well-established M^pro binding site using Glide software, employing the SP scoring function. Furthermore, we also calculated a relative binding affinity ($\Delta G_{bind}$) using the MM/GBSA method. The output of this calculation is reported in Table 1, while the docking results are illustrated in Figure 3.

**Table 1.** Computational analysis regarding compounds **4–7** as potential M^pro inhibitors, along with reference compounds **2** and **3** (N3).

| Compound | Docking Score (kcal/mol) | $\Delta G_{bind}$ (kcal/mol) | QPlogP [B] | QPlogS [C] | PAINS [D] |
|---|---|---|---|---|---|
| **4** | −10.779 | −123.15 | 1.82 | −3.68 | No |
| **5** | −10.027 | −109.41 | 2.32 | −4.68 | No |
| **6** | −11.269 | −114.26 | 3.11 | −4.70 | No |
| **7** | −9.540 | −114.04 | 1.72 | −3.54 | No |

**Table 1.** *Cont.*

| Compound | Docking Score (kcal/mol) | $\Delta G_{bind}$ (kcal/mol) | QPlogP [B] | QPlogS [C] | PAINS [D] |
|---|---|---|---|---|---|
| **2** | −9.976 | −110.55 | 3.23 | −6.04 | No |
| **3**, N3 | −10.138 | −108.36 | 3.18 | −7.25 | No |

[A] QPlogP predicted octanol/water partition coefficient (range or recommended value for 95% of known drugs −2–6.5); [B] QPlogS predicted aqueous solubility in mol/dm$^3$ (range or recommended value for 95% of known drugs: −6.5–0.5); PAINS assessment was performed by FAFDrugs4 online server (accessed on 25 February 2022).

Based on docking results, we observed a significant binding affinity of the developed compounds for the selected target. Considering the retrieved binding mode, we observed that compound **4** (Table 1 and Figure 3A) spanned and interacted with all regions S1–S4 of the M$^{pro}$ binding site (S1–S4). In fact, the difluorostatone moiety established a polar contact with the backbone of G143 (S1′ region), and the pyrrolidinone moiety strongly targeted residues belonging to the S1 region, establishing a series of H-bonds with the backbone of F140 and the sidechains of E166 and H163. The central region of the molecule targeted the backbone of H164 and E166. The P1-moiety of the peptide-based derivative **4** formed H-bonds with the backbone of E166 and with the sidechain of Q192 (S4 region).

The detailed binding mode of compound **4**, represented by a ligand interaction diagram, is visible in Figure 4A. This binding mode accounted for a docking score of −10.779 kcal/mol and a $\Delta G_{bind}$ of −123.15 kcal/mol.

The introduction of the oxazole moiety to replace the methyl group of compound **4** led to the peptide-based derivative **5**. As observed for its parent molecule, it can establish a strong H-bond network within the active site of the enzyme (Figures 3B and 4B). Compound **5** could establish interactions with the backbone of G143 and C145 by its difluorostatone portion, and the pyrrolidinone moiety could strongly target F140, E166, and H163, establishing the same contacts described for compound **4**. The oxazole moiety did not form polar contacts, while it established hydrophobic interactions within the S4 region of the enzyme. This binding mode accounted for a docking score comparable to that found for derivative **4** (GlideScore compound **5**: −10.027 kcal/mol; $\Delta G_{bind}$ −109.41 kcal/mol). A further modification of the tail of compound **5** by introducing a quinoline group aimed at maximizing the number of contacts within the S4 region of the binding site led to the design of the peptidomimetic **6**. The results of the modeling study on derivative **6** are depicted in Figures 3C and 5A. Gratifyingly, our hypothesis was confirmed by the molecular docking calculation. In effect, in addition to the contacts previously described for compound **5**, compound **6** could target the backbone of T190, increasing the hydrophobic contacts within the S4 region. This binding mode, with an improved number of contacts, accounts for a docking score of −11.269 kcal/mol and a $\Delta G_{bind}$ of −114.26 kcal/mol.

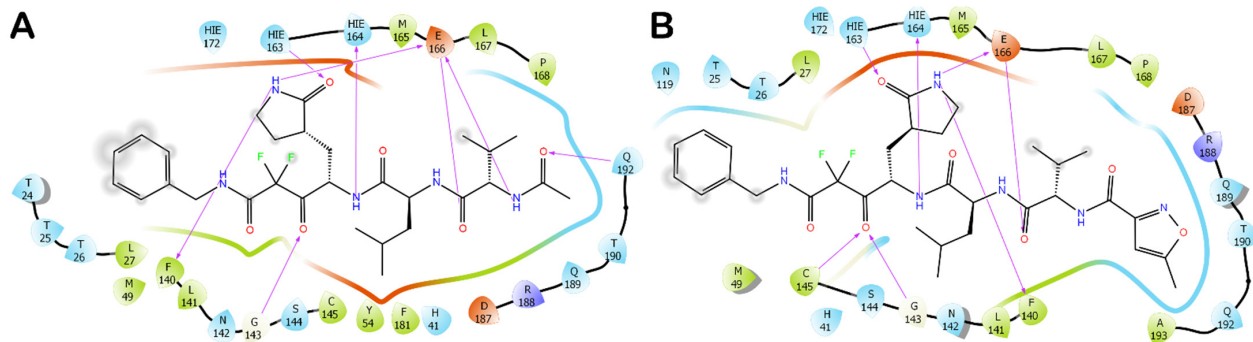

**Figure 3.** Docked pose of compounds **4**–**7** (panels **A**–**D**, respectively) into M$^{pro}$-SARS-CoV-2 (PDB ID: 6Y2G). Key interacting residues from different regions are represented by sticks and labeled. H-bonds are represented as yellow dotted lines. Pictures were generated by Maestro (Maestro, Schrödinger LLC, release 2020-3).

**Figure 4.** Detailed binding modes of compound **4** (panel **A**) and compound **5** (panel **B**). Pictures were generated by ligand interaction diagram available in Maestro (Maestro, Schrödinger LLC, release 2020-3).

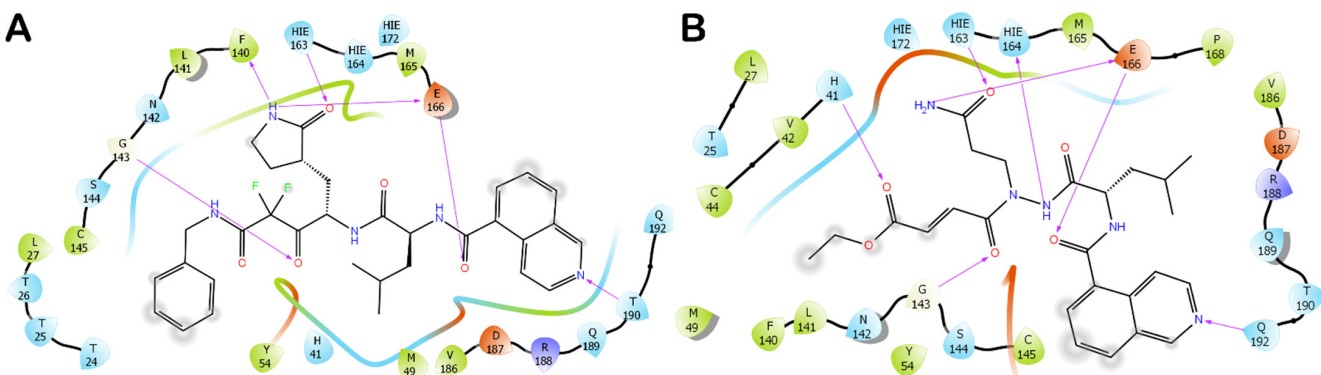

**Figure 5.** Detailed binding mode of compound **6** (panel **A**) and compound **7** (panel **B**). Pictures were generated by ligand interaction diagram available in Maestro (Maestro, Schrödinger LLC, release 2020-3).

Finally, we attempted to modify the head of the molecule by introducing a Michael acceptor and replacing the pyrrolidinone group with glutamine to evaluate a different warhead in the quinoline-based derivative. Interestingly, the resulting compound, derivative **7**, showed a comparable binding mode with respect to the previously discussed molecules. As reported in Figures 3D and 5B, compound **7** could establish the same above-described interactions at the S1 and S1′ regions of the enzyme, targeting G143, H163, H164, and E166. Additionally, we observed an H-bond with the sidechain of H141. Notably, the quinoline moiety at the S4 site established an H-bond with the sidechain of Q192. Although there was a slight decrease in the docking score (−9.540 kcal/mol), the binding affinity (ΔG$_{bind}$ of −114.04 kcal/mol) is in line with the values estimated for the discussed derivatives **4**–**6**. Accordingly, the docking studies confirmed the potential of the selected peptide-based derivatives to target SARS-CoV-2 M$^{pro}$.

Because of the mechanism of the enzyme, it is crucial to evaluate the distance between the reactive residues of the enzyme and the possible atoms of the compound susceptible to the attack for covalent bonding. In particular, the M$^{pro}$ C145 residue represents the pivotal residue to form a covalent adduct. Therefore, we measured the distance between the sulfur atom of C145 and the carbon atom of the compound susceptible to the nucleophilic attack. As reported in Figure S1, the measured distances are for all compounds under 3 Å (compound **4**: −2.87 Å; compound **5**: −2.69 Å; compound **6**: −2.84 Å; compound **7**: −2.93 Å). Remarkably, the findings agree with the possibility that these compounds can form a covalent adduct within the active site of the enzyme, precluding its function.

To compare the mentioned results, we performed further docking calculations, using the same computational protocol, employing two reference compounds, **2** and **3** (N3) (Table 1). According to the crystal structures of the reference compounds, the docking protocol was able to correctly accommodate these ligands within the M$^{pro}$ binding site (Figure S2). Furthermore, these calculations also provided computational scores (Table 1) that can be compared to those obtained for compounds **4**–**7** (Table 1). The analysis of docking scores and ΔG$_{bind}$ indicated that our compounds can bind the M$^{pro}$ binding site with affinities comparable to those observed for the reference compounds **2** and **3**, establishing similar contacts, targeting crucial residues for enzyme activity. Moreover, as reported in Figure S3, as expected, also for reference compounds, the distance between the reactive residues of the enzyme (C145) and the possible atom of the compounds susceptible to the attack for a covalent bonding was found to be compatible with the formation of a covalent adduct within the active site of the enzyme, in line with the experimental activity of reference compounds.

### 3.2. Molecular Dynamics Simulations

After docking studies, we validated the retrieved binding modes by conducting MD simulations in the explicit solvent. We employed $M^{pro}$/ligands docking-derived complexes to investigate the evolution of biological systems for 100 ns. The resultant MD trajectories for all complexes were deeply examined through several standard simulation parameters, including RMSD analysis for all backbone atoms and ligands and RMSF of individual amino acid residue. The selected complexes displayed reasonable stability from the early stages of the simulation, as indicated by observing the RMSD. Considering the entire simulation time, we did not observe any major expansion and/or contraction, caused by the binding of the investigated compounds (Figure 6, regarding the simulation of compounds **4**–**7**). This stability was also corroborated by examining the RMSF determined for the selected complexes. RMSF denotes the variation between the atomic $C\alpha$ coordinates of the enzyme from its average position during the MD simulation. This computation is profitable to characterize the flexibility of individual residues of the protein backbone. The systems under study did not show considerable fluctuation events, with the exclusion of an extremely limited number of residues at the N- and C-terminal regions of $M^{pro}$ (Figure S4). Likewise, the conformational adaptations of critical residues in the active site (lowest RMSF values for all complexes) confirmed the ability of compounds to form stable interactions within the protein.

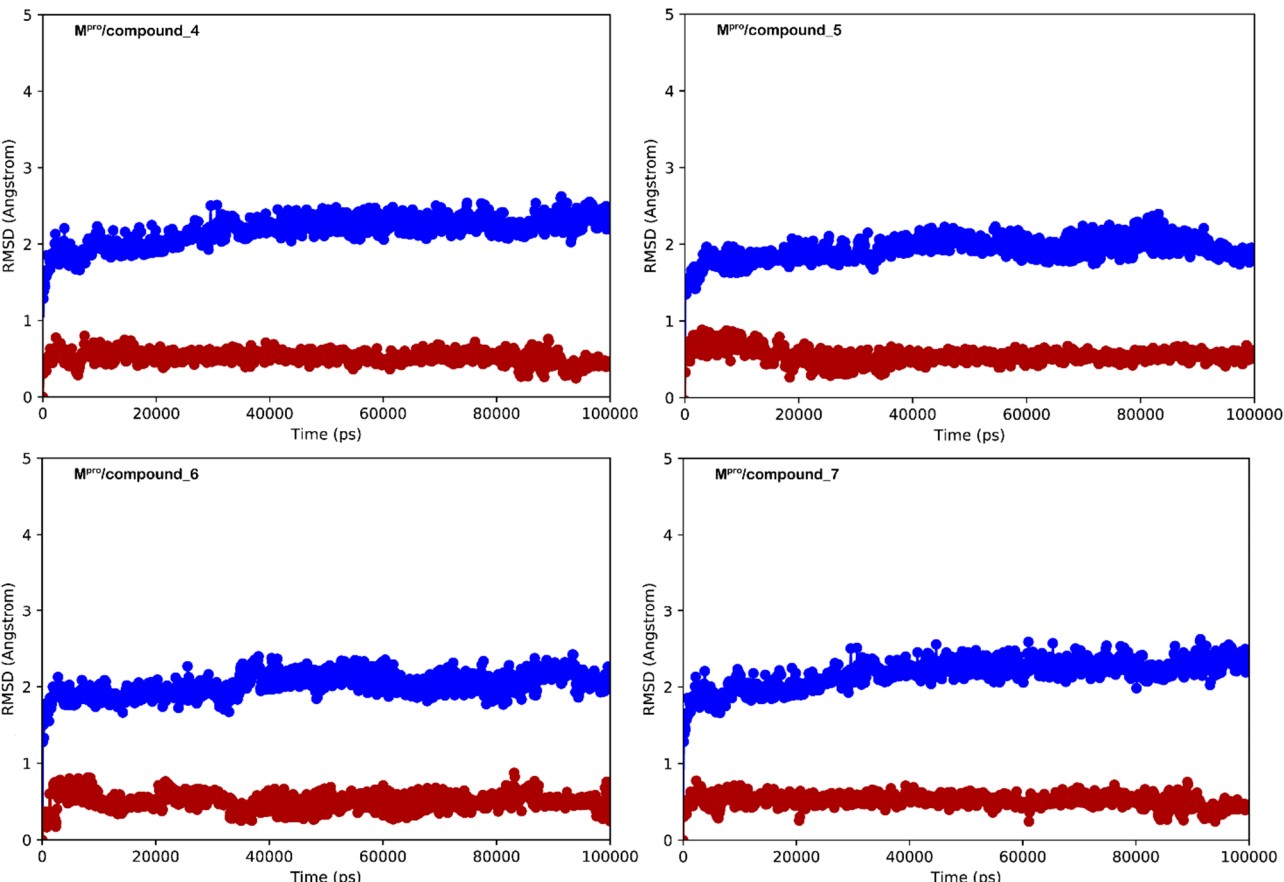

**Figure 6.** RMSD calculation for each complex investigated in this study: protein (blue line) and ligand (red line). Pictures were generated by Maestro (Maestro, Schrödinger LLC, release 2020-3).

To better comprehend the behavior of derivatives **4**–**7** into the SARS-CoV-2 $M^{pro}$ binding site, we performed a comprehensive analysis of MD simulations, exploring the established contacts. In general, compound **4** (Figures 7A and S5A) maintained the contacts found by docking calculation. Interestingly, we observed a stronger network of interactions

at the S1′ region since contacts with S144 and the backbone of C145 were detectable. Furthermore, the tail of compound **4**, in addition to the H-bond with Q192, established additional H-bonds with Q189 and T190 (S3 region), sometimes water-mediated. The analysis conducted on the trajectory of MD simulation for compound **5** is illustrated in Figures 7B and S5B. Here again, the crucial contacts established by compound **5** within the M$^{pro}$ binding site were maintained. Notably, the strong network of contacts at the S1 and S1′ regions was conserved during the simulation with the addition of contacts with H41. The tail becomes able to effectively target Q189, T190, and Q192 at the S3 and S4 regions, resulting in a more tightly binding within the active site of the enzyme.

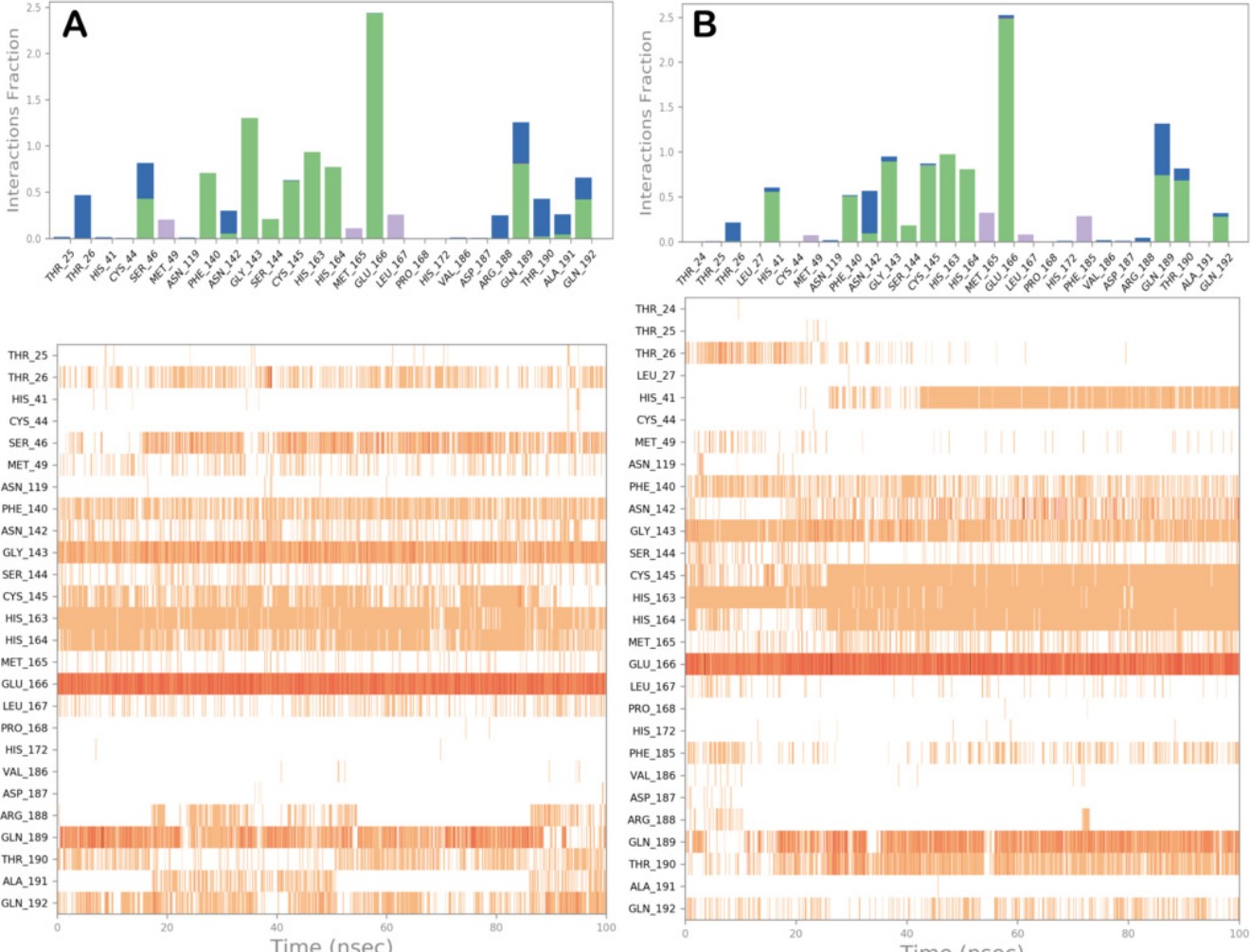

**Figure 7.** Compounds **4** (panel **A**) and **5** (panel **B**) monitored during the simulation. The contacts can be grouped by type and summarized, as shown in the plots. Grouping protein–ligand interactions into four types: H-bonds (green), hydrophobic (gray), ionic (magenta), and water bridges (blue). In the second graph of the picture is reported a timeline representation of the contacts. Some residues make more than one specific contact with the ligand, which is represented by a darker shade of orange. Pictures were generated by the simulation interaction diagram available in Desmond via Maestro (Maestro, Schrödinger LLC, release 2020-3).

Additionally, the MD analysis of compound **6** (Figures 8A and S5C) and compound **7** (Figures 8B and S5D) revealed a similar trend. Regarding compound **6**, interactions with H41 at the S2 site and Q189 and Q192 at site S3 in addition to the contacts found by molecular docking studies were observed. Compound **7** showed a comparable behavior since it can strengthen the interaction within the active site, establishing a strong network

of contacts at S3 with Q189 and T190 and increasing the contacts with Q192. Overall, the MD simulation analysis indicated a high stability of the binding mode found by molecular docking for each selected complex. In addition to the existing contacts, we found a reasonable number of novel contacts that can further stabilize the binding modes.

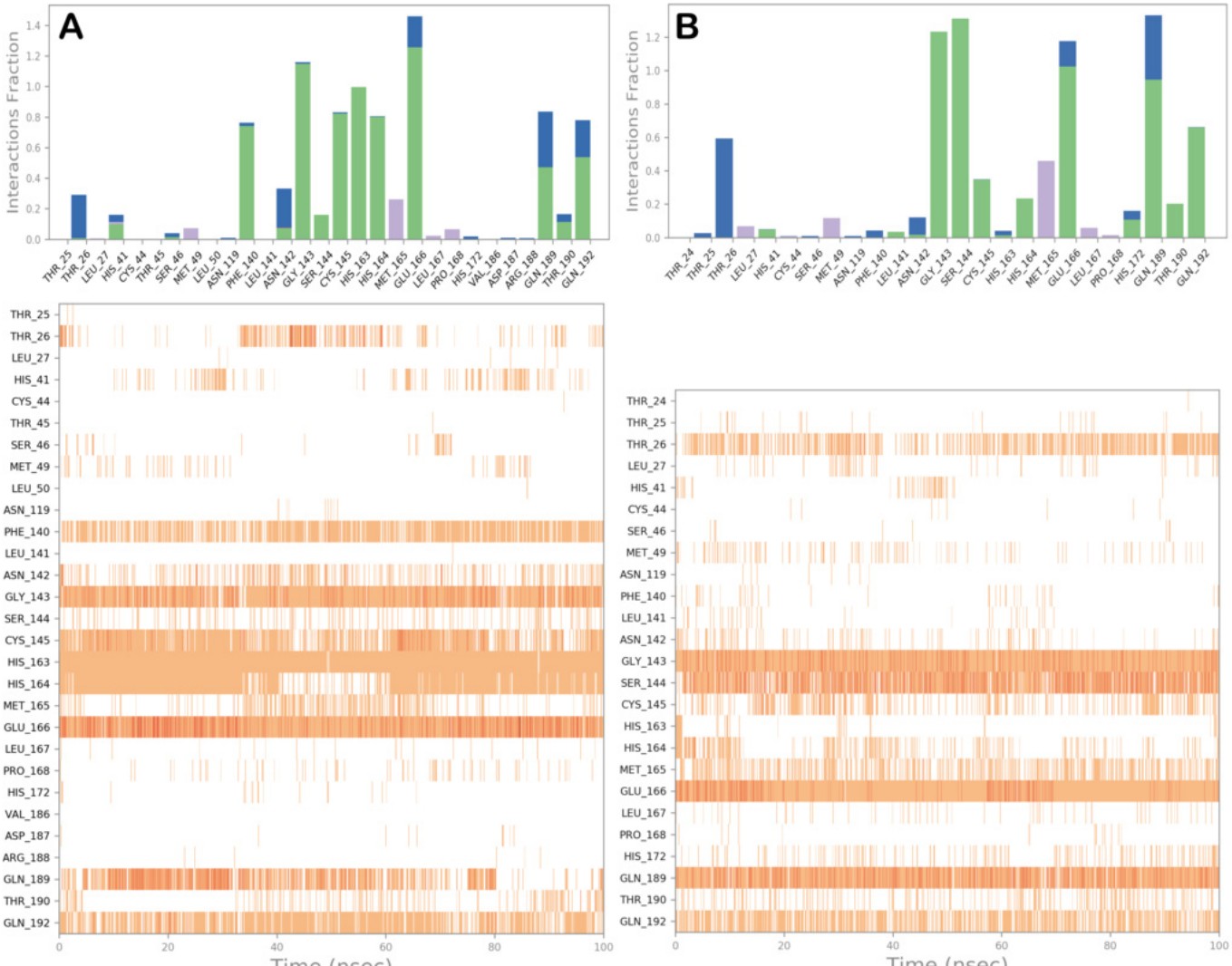

**Figure 8.** Compounds **6** (panel **A**) and **7** (panel **B**) monitored during the simulation. The contacts can be grouped by type and summarized, as shown in the plots. Grouping protein–ligand interactions into four types: H-bonds (green), hydrophobic (gray), ionic (magenta), and water bridges (blue). In the second graph of the picture is reported a timeline representation of the contacts. Some residues make more than one specific contact with the ligand, which is represented by a darker shade of orange. Pictures were generated by the simulation interaction diagram available in Desmond via Maestro (Maestro, Schrödinger LLC, release 2020-3).

We then monitored the distance between the sulfur atom of C145 and the electrophilic carbon atom of the ligand, susceptible to nucleophilic attack for each complex. As reported in Figure S6, the distance between the selected atoms remained mainly constant with very small variations, as expected due to the high stability of the complexes. Accordingly, the measures indicated that the considered electrophilic carbon atom remained susceptible to a possible nucleophilic attack from C145 during the simulation time.

In addition, to further validate our computational protocol, we performed MD simulations also for the ligand/enzyme complexes of the reference compounds previously described (Figure S2). As observed for compounds **4**–**7**, the investigated systems were

reasonably stable with small fluctuations (Figure S7), and the contacts found by docking studies were maintained during the MD simulations (Figure S8). As expected, the distances between the reactive residues of the enzyme (C145) and the possible atom of the compounds susceptible to the attack for a covalent bonding were also found to be stable during the simulations (Figure S9), indicating the reliability of the computational approach.

Finally, to further corroborate the obtained results, we performed additional calculations, using the FEP technique to compute the differences in protein–ligand-binding free energies from MD simulations. The output of this calculation in terms of $\Delta\Delta G_{bind}$ is reported in Table 2, with compound **2** (crystallized ligand in the structure 6Y2G, used in this study) employed as the reference compound. As indicated by the results, compounds **4–6** showed an improved binding affinity with respect to the reference molecule (compound **2**), while compound **7** showed a slight decrease in binding affinity consistent with lower computational scores, found by other methods, with respect to the best performing compounds. Notably, FEP calculation confirms the potency of compound **3** in inhibiting M$^{pro}$ with a slight improvement with respect to the value found for the 6Y2G ligand (compound **2**) [8,12].

**Table 2.** Computational scores (covalent docking score and $\Delta G_{bind}$ derived from docking studies, and $\Delta\Delta G_{bind}$ derived by FEP calculation) obtained for compounds **4–7** compared to the reference compounds **2** and **3**.

| Compound | Covalent Docking Score (kcal/mol) | Covalent Docking $\Delta G_{bind}$ (kcal/mol) | FEP/MD $\Delta\Delta G_{bind}$ (kcal/mol) |
|---|---|---|---|
| 4 | −10.834 | −128.29 | −0.18 ± 0.11 |
| 5 | −10.232 | −119.17 | −0.45 ± 0.21 |
| 6 | −11.681 | −116.49 | −0.73 ± 0.32 |
| 7 | −9.828 | −115.96 | 0.12 ± 0.09 |
| 2 | −10.174 | −113.87 | – |
| 3,N3 | −10.043 | −114.74 | −0.13 ± 0.12 |

### 3.3. Covalent Docking Approach

To gain further insight into the formation of the tetrahedral intermediate and predict the binding mode of peptide-based derivatives, different molecular models of the selected complexes were generated using a covalent docking protocol, namely CovDock, available in Maestro. Once the correct reaction is written and the software recognizes all the residues involved, CovDock initially combines the Glide docking algorithm and Prime structure refinement to determine whether the ligand can be accommodated into the selected binding site (standard docking). In this way, as a constraint, the ligand should sit in a position close enough to the nucleophilic group of the reactive residue. The reactive residue, cysteine, is mutated with an alanine residue to generate an initial association in which the ligand is noncovalently bound to the target protein. Subsequently, the receptor is restored, and the reaction occurs. Once the covalent bond is formed, the complex is minimized. Now, the obtained poses are clustered and ranked after a complete minimization. The output of this calculation is illustrated in Figure 9. The docked poses of compounds **4–7** into the catalytic site of M$^{pro}$ were chosen as the starting point for the covalent docking procedure. As displayed in Figure 9, the tetrahedral intermediates can be stabilized by the formation of novel H-bonds with specific amino acid residues, thus resulting in an overall fine-tuning of the binding conformation of compounds **4–7** within the binding cleft and in the generation of more stable complexes. Accordingly, based on this computational approach, the conceived compounds can act as covalent inhibitors of SARS-CoV-2 M$^{pro}$.

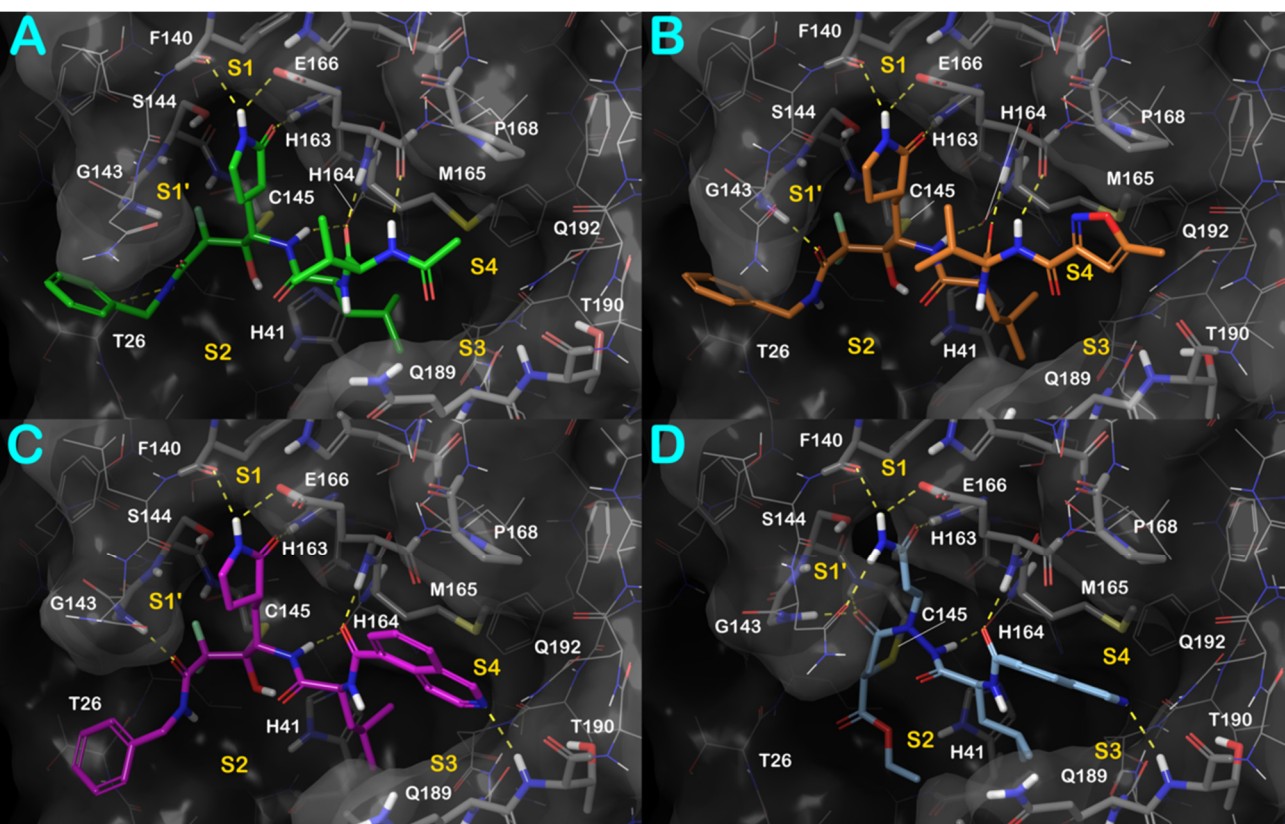

**Figure 9.** Output regarding the covalent docking investigation considering compounds **4**–**7** (panels **A**–**D**, respectively). Key interacting residues from different regions are represented by sticks and labeled. H-bonds are represented as yellow dotted lines. Pictures were generated by Maestro (Maestro, Schrödinger LLC, release 2020-3).

In this case, we conducted the same computational study on reference compounds. The adopted covalent docking protocol is effectively able to correctly accommodate the reference compounds within the M$^{pro}$ binding site with high accuracy, reproducing the crystal structure conformation of reference compounds when they are covalently bound to the binding site (Figure S10). Remarkably, the geometry obtained for the reference ligands covalently bound to M$^{pro}$ is very close to that observed in the crystal structures [8,12]. Gratifyingly, the computational docking scores reported in Table 2 further confirmed the susceptibility of compounds **4**–**7** to react within the M$^{pro}$ binding site, forming a covalent adduct with C145 due to the comparable scores found for the reference compounds.

## 4. Conclusions

In summary, we have described a computational protocol aimed at designing novel SARS-CoV-2 M$^{pro}$ covalent inhibitors. The work was focused on the evaluation of bifunctional warheads engaging prime and nonprime subsites of the active site of the enzyme. To this end, we designed, considering the binding site of M$^{pro}$, peptide-based inhibitors based on the difluorostatone scaffold that has been demonstrated to be effective in inhibiting other proteases [13–15]. In addition, a peptide-based inhibitor containing a Michael acceptor has been designed. All these compounds were computationally investigated using several in silico techniques such as molecular docking, covalent docking, MD simulation, and FEP, for evaluating their potential as covalent inhibitors against SARS-CoV-2 M$^{pro}$. Computational hints indicated that the proposed compounds can be effective in inhibiting the enzyme, deserving further experimental studies to confirm these findings to expand the armamentarium for fighting this virus. Moreover, our work provides a rational computer-driven

approach for developing covalent inhibitors of the M$^{pro}$ enzyme. This approach could also be extended to the inhibition of other drug targets.

**Supplementary Materials:** The following supporting information can be downloaded at: https://www.mdpi.com/article/10.3390/computation10050069/s1, Figure S1: Measured distances between the sulfur of the catalytic residue C145 and the electrophilic carbon of compounds **4** (panel A), **5** (panel B), **6** (panel C), and **7** (panel D) that can be susceptible of nucleophilic attack; Figure S2: Docked pose of compound **2** and compound **3** (N3) (panels A,B, respectively) into M$^{pro}$-SARS-CoV-2 (PDB ID: 6Y2G); Figure S3: Measured distances between the sulfur of the catalytic residue C145 and the electrophilic carbon of compound **2** (panel A) and compound **3** (N3) (panel B) that can be susceptible to nucleophilic attack; Figure S4: RMSF calculation for each complex, selected by docking studies, after 100 ns of MD simulation; Figure S5: Dynamic ligand interaction diagram regarding compounds **4** (panel A), **5** (panel B), **6** (panel C), and **7** (panel D), calculated through 100 ns of MD simulation; Figure S6: Monitored distances between the sulfur of the catalytic residue C145 and the electrophilic carbon of compounds **4** (panel A), **5** (panel B), **6** (panel C), and **7** (panel D) that can be susceptible to nucleophilic attack; Figure S7: RMSD calculation for each complex investigated in this study: protein (blue line) and ligand (red line); RMSF calculation for each complex, selected by docking studies, after 100 ns of MD simulation (panel A, compound **2**; panel B, compound **3** (N3)); Figure S8: Compound **2** (panel A) and compound **3** (N3) (panel B) monitored during the simulation; Figure S9: Monitored distances between the sulfur of the catalytic residue C145 and the electrophilic carbon of compound **2** (panel A) and compound **3** (N3) (panel B) that can be susceptible to nucleophilic attack; Figure S10: Output regarding the covalent docking investigation considering compound **2** (panel A) and compound **3** (N3) (panel B).

**Author Contributions:** Conceptualization, S.B. (Simone Brogi) and S.G.; methodology, S.B. (Simone Brogi), R.I., S.B. (Stefania Butini) and S.G.; software, S.B. (Simone Brogi); validation, S.B. (Simone Brogi), S.B. (Stefania Butini), V.C., G.C. and S.G.; investigation, S.B. (Simone Brogi), S.R., S.B. (Stefania Butini) and S.G.; data curation, S.B. (Simone Brogi), S.R., R.I., S.B. (Stefania Butini), V.C., G.C. and S.G.; writing—original draft preparation, S.B. (Simone Brogi) and S.G.; writing—review and editing, S.B. (Simone Brogi), S.R., R.I., S.B. (Stefania Butini), V.C., G.C. and S.G.; supervision, S.B. (Simone Brogi) and S.G.; funding acquisition, S.B. (Simone Brogi) and S.G. All authors have read and agreed to the published version of the manuscript.

**Funding:** This research was funded by "Fondo di Beneficenza di Intesa Sanpaolo" (Grant Number B/2020/0113 to S.B. (Simone Brogi) and S.G.).

**Institutional Review Board Statement:** Not applicable.

**Informed Consent Statement:** Not applicable.

**Data Availability Statement:** Not applicable.

**Conflicts of Interest:** The authors declare no conflict of interest.

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
