# Peer review of "In Silico Analysis of Peptide-Based Derivatives Containing Bifunctional Warheads Engaging Prime and Non-Prime Subsites to Covalent Binding SARS-CoV-2 Main Protease (Mpro)"

_computation, doi:10.3390/computation10050069_

Round 1

Reviewer 1 Report

The research presented in the manuscript is interesting and urgently needs to be published. It covers the most advanced methods in computational medicinal chemistry and addresses current global problems. Computation could be the perfect host for this article.

Providing information when the link https://fafdrugs4.rpbs.univ‐paris‐diderot.fr/ was accessed in line 203 will make the manuscript better.

Author Response

The research presented in the manuscript is interesting and urgently needs to be published. It covers the most advanced methods in computational medicinal chemistry and addresses current global problems. Computation could be the perfect host for this article.

Authors: we thank the referee for the positive evaluation of the work and for the appreciation of the manuscript.

Providing information when the link https://fafdrugs4.rpbs.univ‐paris‐diderot.fr/ was accessed in line 203 will make the manuscript better.

Authors: According to the referee’s suggestion we reported the accessed date to the webserver.

Reviewer 2 Report

The authors report a standard application of the Schrödinger suite with a focus on screening compounds against the Mpro target. There is nothing particularly wrong with the paper, though it is close to a tutorial-like Schrödinger exercise rather than a manuscript of top-novelty.

In spite of such limitation, this referee feels the paper might fit to what one expects for a contribution in Computation, as the numeric outcomes might be of interest in the design of antivirals with application in COVID-19.

However, I have one major correction that must be correctly addressed prior publication:

1. Authors must prove that their computational protocol is able to mimic the biological activity of available Michael-based Mpro inhibitors. More specifically, I am asking for a correlation of binding energies vs. IC50 values. It is critical to ensure that the predicted inhibition by theory, which is based on values listed in Table 1, is able to reproduce measured values in the laboratory.

I will recommend this paper for publications only after benchmarking that standard Schrödinger protocol.

General comment:

2. More advanced levels of theory are implemented in that same suite, e.g., FEP. Such additional calculations will increase the quality of the present manuscript.

Author Response

The authors report a standard application of the Schrödinger suite with a focus on screening compounds against the Mpro target. There is nothing particularly wrong with the paper, though it is close to a tutorial-like Schrödinger exercise rather than a manuscript of top-novelty.

In spite of such limitation, this referee feels the paper might fit to what one expects for a contribution in Computation, as the numeric outcomes might be of interest in the design of antivirals with application in COVID-19.

Authors: we thank the referee for the positive evaluation of the work and for the appreciation of the manuscript, and for the suggestions that allowed us to improve the quality of the manuscript and the reliability of the proposed results.

However, I have one major correction that must be correctly addressed prior publication:

  1. Authors must prove that their computational protocol is able to mimic the biological activity of available Michael-based Mpro inhibitors. More specifically, I am asking for a correlation of binding energies vs. IC50 values. It is critical to ensure that the predicted inhibition by theory, which is based on values listed in Table 1, is able to reproduce measured values in the laboratory.

Authors: the observation raised by the reviewer is of high interest. As the referee know, the correlation of biological activity with computational scores is a challenging task. Moreover, binding energies should be better correlated with binding affinity such as Ki, Kd and so on, rather than IC50. In fact, the inhibitory concentration is an index of the potency of compounds and should not be directly correlated with the affinity. However, for better circumstantiate the findings on compound 4-7, we reported the data regarding two reference inhibitors (compounds 2 (ligand belonging to the 6Y2G structure) and 3 (N3)). In fact, we performed a comparison adopting the same protocol (molecular docking, MD simulation, and covalent docking) among our ligands and reference compounds. We added a discussion in the main text and several pictorial representations about these calculations in the Supplementary Materials file.

General comment:

  1. More advanced levels of theory are implemented in that same suite, e.g., FEP. Such additional calculations will increase the quality of the present manuscript.

Authors: according to the suggestion, for increasing the quality of the manuscript, we performed a FEP calculation considering our ligands and the reference inhibitors. In particular, using the 6Y2G ligand (compound 2) as reference compound in the calculation, FEP results in term of ΔΔGbind highlighted the potential of our designed compounds. We added a discussion regarding these results before Table 2 (table added in the revised version with the ΔΔGbind values).

Round 2

Reviewer 2 Report

Authors have correctly addressed my main comment, e.g., the use of a higher level of theory to confirm the results provided in the original version of the manuscript.

I have noted hew text in page 14: 'Notably, FEP calculation confirm the potency of compound 3 in inhibiting Mpro 392 with a slight improvement with respect to the value found for the 6Y2G ligand'.

I have also inspected new numeric values listed in Table 2. It is satisfying that my suggestion was used to further improve the quality of the paper, which a I feels is now suitable for publication without any other change.

Congratulations to all authors for such nice example of computational work.

Author Response

We wish to thank the referee for the appreciation of the revision done